# A Review of Spatial Mismatch Research: Empirical Debate, Theoretical Evolution and Connotation Expansion

**Liping Wang [1,2], Cifang Wu [3] and Songnian Zhao [4,\*]**

1.  Law School, Shantou University, Shantou 515063, China; wanglp@stu.edu.cn
2.  Local Government Development Research Institute, Shantou University, Shantou 515063, China
3.  Land Academy for National Development, Zhejiang University, Hangzhou 310029, China; wucifang@zju.edu.cn
4.  School of Government, Shenzhen University, Shenzhen 215123, China
\*   Correspondence: zhaosn@szu.edu.cn

**Abstract:** At present, widespread urban expansion, regeneration, and transformation have inevitably led to the spatial separation of residence and employment, and negatively affected the employment welfare of the subjects concerned, which needs to be traced back to the theory of spatial mismatch to explore possible solutions. The theory of spatial mismatch has been proposed for more than sixty years, and its theoretical connotation has been continuously expanded through the heated debate on its objective existence. However, due to the lack of understanding of its expansion process and the essence of its theoretical connotation, its theoretical meaning is ambiguous and fails to truly guide its role in practice. Based on the structural analysis of the connotation of spatial mismatch, this study summarizes the theoretical evolution and empirical development of spatial mismatch from four levels: "objects of concern-influencing factors-spatial relations-consequential effects". It is found that after half a century of evolution, the study of spatial mismatch has formed a relatively perfect theoretical and methodological system, and in the process of expansion, spatial mismatch has been given a deeper meaning, which can provide an important theoretical and practical reference for solving the separation of residential and spatial resources and the resultant welfare loss effects.

**Keywords:** spatial mismatch; empirical debates; theoretical evolution; connotation expansion

## 1. Introduction

The transformation and development of territorial and urban systems, whether it is incremental expansion or spatial reconfiguration of the stock, inevitably brings about the restructuring of urban industries and land use, and in turn brings about a trend of spatial separation of residence and employment [1,2]. This separation is considered to be the main cause of the employment and income difficulties of some groups, and this dilemma is even more acute for the disadvantaged groups having explicit barriers to mobility and access to the separated employment resources [3]. How to deal with the residence-employment separation and the consequential employment difficulties has been identified as goals for cities and communities. In this regard, there is an urgent need to trace theories and find possible solutions.

The theoretical basis for the study of residential-employment spatial separation and the resulting job losses can be traced back to the spatial mismatch hypothesis developed in the late 1960s [4]. At that time, technological innovation, rising land prices in urban centers, and declining transportation costs combined to bring about a massive and sustained suburbanization of metropolitan America. While the rapid growth in labor demand (particularly in manufacturing and retail for low-skilled, blue-collar workers) has shifted from the inner city to the suburbs, Black ethnicity has been passively segregated in the inner city due to suburban housing market discrimination, or actively segregated in the inner city due to "residential stickiness", resulting in a spatial separation of Black people from suitable jobs

in the suburbs, and then a decline in income and even unemployment [5,6]. In Kain's article "Residential Segregation, Black Employment, and Metropolitan Dispersal", Kain found and tested that "the spatial separation of inner-city black neighborhoods from suburban jobs in the Chicago and Los Angeles metropolitan areas induces high unemployment rates among inner-city blacks" [4]. This finding was then refined into the spatial mismatch hypothesis and its implications were clearly defined: the subject concerned is on inner-city Blacks, the influencing factors are the suburbanization of employment in metropolitan America and racial discrimination in the housing market, and the relationship manifests itself as a spatial separation of residence and employment, resulting in high unemployment, low incomes, and long commutes for inner-city Blacks.

Soon after the spatial mismatch hypothesis was put forward, empirical studies using different indicators, methods, and studies on different groups have intensively debated the objective existence of spatial mismatch [5–8]. With the continuous expansion of spatial mismatch research in terms of groups concerned, influencing factors, and spatial relationship metrics, more and more scholars have moved from objective existence examination to asking about the mechanisms of spatial mismatch [9–12], and have devoted themselves to extracting and summarizing from empirical studies and constructing theoretical models to upgrade spatial mismatch hypotheses into theories [13–18]. Meanwhile, scholars in various countries have combined spatial mismatch research with country-specific scenarios such as public housing allocation, public transport development, urban spatial planning, and slum upgrading, making spatial mismatch theory widely used [19–24]. In a word, spatial mismatch has been studied for more than half a century, and has gone through different stages of theoretical hypothesis, objective existence debate, and hypothesis rising into theory; its connotation has been enriched and its application has been expanded. However, there are few studies to sort out the whole process of the development of spatial mismatch theory and to refine its theoretical essence, making the current research on spatial mismatch biased in the following two aspects: one is to confuse spatial mismatch with housing segregation, job accessibility, and job-housing separation, and the other is to focus on the analysis of the spatial structural relationship between housing and employment at the macro level of aggregation, ignoring the impact on the relevant groups in terms of employment participation, wage income, and other labor market dilemmas. As a result, most of the current studies point to macro spatial distribution issues such as the balance of employment and residence, as well as urban planning, and lack sociological perspectives on issues such as social justice and spatial equity, which deviates from the original purpose and essence of spatial mismatch theory.

Many urban spaces in the developing world, represented by China, are currently in a period of rapid sprawl. The resulting urban spatial alienation and separation of residence and employment is a phenomenon that has become obvious. The disadvantaged groups who cannot afford the high housing prices in the city center and are gradually moving to the urban periphery are facing similar spatial barriers to employment and welfare deprivation as the Blacks in the inner cities of the United States, and failure to address these problems in time will likely lead to spatial injustice and social imbalance [1,25]. In the face of this emerging reality, it is necessary to clarify a series of questions such as "which groups are affected by spatial mismatch, what are the manifestations of spatial separation, to what extent does the separation lead to welfare loss for the groups concerned, and what are the real causes of spatial mismatch" in order to find answers to these issues. This requires a return to the theory of spatial mismatch itself and a reconsideration of the practical and theoretical propositions that it implies. In fact, since its first formulation by Kain, the spatial mismatch hypothesis has illustrated the theory elements, including the group concerned, influencing factors, spatial relationship manifestation, and consequential effects hierarchically. Therefore, this study aims to analyze the hierarchical elements of the spatial mismatch theory, and to sort out the whole process from the debate to the continuous expansion of its theoretical elements, as well as to reaffirm its theoretical value in solving the increasingly prominent spatial and social injustice of many cities.

## 2. The Debate on the Objective Existence of Spatial Mismatch

Once the spatial mismatch hypothesis was formulated, studies were conducted to examine the validity and generalizability of the hypothesis using empirical statistical methods. A number of studies revolved around several levels of the initial definition of spatial mismatch, seeking to falsify it in terms of the limitations of the indicator methods used, the non-dominant influence of spatial distance, and the fact that the subjects concerned were not restricted to inner-city Blacks. Many studies, on the other hand, have more or less confirmed spatial mismatch along Kain's lines, and since the 1990s there have been a number of reviews that support spatial mismatch, with a long-running debate between the verifying and disproving sides over the objective existence of spatial mismatch.

### 2.1. Falsification of Spatial Mismatch

Although Kain's research on the relationship between residential segregation of Blacks and their employment has pioneered a new research direction, his study has been questioned since its inception, with some scholars questioning the objective existence of spatial mismatch from the perspectives of the validity of the selection of indicators and the scientific nature of the model construction, e.g., the Quarterly Journal of Economics published three consecutive papers discussing the rationality of his research methodology and the credibility of his findings with Kain [26–28]. At the beginning of the second half of the 1980s, more studies questioned the objective existence of spatial mismatch in terms of the importance of spatial distance, arguing that the spatial barriers to employment resulting from the geospatial separation of residence and employment had a minimal effect, and that racial mismatch [7], transportation mismatch [29], skill mismatch [30], information mismatch [31], and even policy mismatch [32] mainly account for the employment loss of disadvantage Blacks (Table 1). Other studies have found that suburban areas may face similar spatial mismatch problems as inner cities [33], and that Whites trapped in inner cities for economic reasons also are affected by spatial mismatch [8], suggesting that there are more possibilities for the spatial mismatch hypothesis than just a simple formulation of the initial definition.

**Table 1.** Representative literatures of the various mismatch types.

| Mismatch Type | Representative Literatures | Main Conclusions |
| --- | --- | --- |
| Racial Mismatch | Ellwood (1983) [7] | Barriers to employment for Blacks do not stem from spatial factors, but rather from racial issues arising from skin color |
| Transportation Mismatch | Ong & Miller (2005) [29] | Heavy reliance on the slow public transport is the real reason for the long commute times for residents of inner cities |
| Skill Mismatch | Stoll (2005) [30] | The mismatch between lower-skilled inner-city minorities and traditional lower-skilled jobs in the suburbs could explain a large proportion of the Black-White employment gap |
| Information Mismatch | Parks (2004) [31] | Good social relations in the family can improve employment outcomes for migrants, especially for women |
| Policy Mismatch | Chapple (2006) [32] | Many policy measures to improve the links between minorities and suburban job opportunities have not significantly improved their employment outcomes |

### 2.2. Confirmation of Spatial Mismatch

Most empirical studies still confirm the objective existence of spatial mismatch, especially since the second half of the 1980s when American urban sociologists re-emphasized the theoretical assumption of spatial mismatch in the face of the persistence of poverty clusters and high unemployment rates among inner-city residents. Keith R. Ihlanfeldt was probably the most active researcher on spatial mismatch during this period, publishing several empirical studies on employment suburbanization and the Black-White wage gap [33], employment accessibility and the employment rates of Black youth [34,35], and employment accessibility and the employment earnings of Hispanic youth [36], all of whose results

fit the description of the spatial mismatch hypothesis. Other scholars' findings, controlling for variables such as individual characteristics, family characteristics, neighborhood characteristics, local unemployment rates, and population growth, all confirm to a greater or lesser extent the objective existence of spatial mismatch [37,38]. Most of the review studies since the 1990s have also supported the objective existence of spatial mismatch. Kain sorted out the impact of housing market discrimination on Black employment and wage levels from the perspective of labor economics, and responded to empirical studies questioning spatial mismatch in each case [39,40]. Some scholars have analyzed the results of different or even contradictory empirical studies from a methodological perspective [41,42]. More scholars have found through the review that with the expansion of the object of spatial mismatch research, the maturity of research methods, and the enrichment of influencing factors, the debate about whether spatial mismatch exists objectively is no longer meaningful [43–45].

The continuing debate over the objective existence of spatial mismatch is related to the inconsistent understanding of the connotations of the spatial mismatch hypothesis. As spatial mismatch was initially proposed to emphasize the adverse effects of space and distance on the labor market outcomes of inner-city Blacks, the early connotation of spatial mismatch was relatively clear but narrowly defined: the subject of the study was Black ethnicity, and the main influences on Black employment disadvantage were necessarily derived from spatial factors in the context of residential segregation and employment suburbanization. Studies questioning the objectivity of spatial mismatch have argued that any inconsistency with these four levels of representation falsifies spatial mismatch. Differences in the subjects, influences, and measures of spatial mismatch can lead to arguments about spatial mismatch. It is important to note, however, that while Kain focused on inner-city Black ethnicity at the beginning of the spatial dislocation proposal, he did not deny the spatial mismatch faced by other groups, particularly vulnerable groups. Although Kain emphasized the role of space as well as distance in the labor market outcomes of Black ethnicity, he clearly recognized how space was interconnected and embedded with social structures, racial discrimination, and labor market functioning.

## 3. Expansion of the Research Scope of Spatial Mismatch

A growing number of studies have realized that spatial mismatch itself was from the beginning not only about how space interacts with the labor market, but about how space is related to race, transport, skills, social relations, occupational segmentation, and other elements that influence labor market outcomes, and that spatial mismatch needs to be understood and constructed in a broader context [40,45]. With further suburbanization, changes in demographic characteristics, housing market reforms, and economic restructuring, and urban sprawl, scholars have begun to reassess the diametrically opposed findings on spatial mismatch by placing them in the new socio-economic context and working to expand the possibilities of spatial mismatch in terms of the group concerned, influencing factors, indicator methods, spatial relationships, and consequential effects.

### 3.1. Expansion of the Group Concerned

Black inner-city Americans face employment vulnerability from the separation of residence and employment, as do other minorities in inner cities. Employment suburbanization can also negatively affect the employment rates of inner-city Hispanic and Latino workers [36,38,46], and even inner-city Whites experience employment vulnerability due to the separation of residential-employment space [47]. Youth are more vulnerable to job market fluctuations due to their lack of employability skills and transportation options [48], and the employment participation rate of ethnic minority youth in particular declines significantly due to reduced employment accessibility [35,37]. Female workers also present frequent employment disadvantages due to occupational segmentation and family responsibilities in the process of suburbanization of employment [49]. Other disadvantaged groups, such as immigrants [50], low-skilled [51], low-income groups [52,53], entry-level workers [54],

individuals recently released from prison [55], and welfare-recipients [56], have entered the study of spatial mismatch in America.

Since then, scholars in other countries have found that not only disadvantaged groups, but also ordinary workers may face employment disadvantages arising from spatial mismatch. Groups such as city dwellers, community dwellers, and general commuters have also become the subject to be included in the study of spatial mismatch [57]. However, subsequent studies have argued that there may be considerable differences in the spatial separation of residence-employment and the resulting negative labor market impacts for different groups, i.e., the spatial separation of residence and employment may not necessarily face spatial mismatch, as some groups may rely on transport as well as personal capabilities to compensate for the threats posed by spatial separation. Therefore, disadvantaged groups remain the focus of spatial mismatch research, but only that the types of disadvantaged groups vary over time and depending on their locations. In China, disadvantaged groups living in low-income settlements such as subsidized housing, urban villages, and old neighborhoods are the main targets of spatial mismatch studies [58–60].

*3.2. Complication of Influencing Factors*

Since the late 1990s, spatial mismatch research has shifted from simply examining spatial mismatch to disentangling the influences that trigger it. Early scholars had mainly attributed spatial mismatch in US metropolitan areas to the suburbanization of employment and the residential-employment separation under housing market segregation [4,5]. Later, a growing number of studies have argued that it is important to focus not only on the spatial separation of residence and employment, but also on the processes of connection or barriers between employment opportunities and workers [45,61], i.e., what prevents ethnic minorities such as inner-city Blacks from supplying labor to suburban areas where labor demand is growing. Residential immobility, commuting barriers, and job searching barriers form three major barriers to spatial mismatch: (1) In the context of suburbanization and urban spatial reconstruction, the lack of residential mobility makes it spatially difficult for ethnic minorities, such as Blacks, to move closer to jobs and increases their employment possibilities [62]; (2) The lack of cars and the inaccessibility of public transport in situations where residence is far from employment result in barriers to commuting [36]; (3) Weaker access to employment information and inefficient job searching behavior may exacerbate the employment vulnerability of the groups concerned in the absence of the separation of their residence and employment [63]. The current widespread urban sprawl, urban renewal in many developing country cities, such as the reconstruction of city centers, shantytown renovation, new town development, and the construction of affordable housing in suburbs, have shifted a large number of vulnerable groups to the urban periphery, inevitably leading to the spatial separation of vulnerable groups from the job-rich central areas. Meanwhile, the combination of barriers to migration caused by high property prices in central areas, barriers to commuting caused by poor transport facilities in the suburbs, and barriers to job search caused by a lack of access to employment information leads to spatial mismatch [1,58,64].

Therefore, the research which were originally presented to falsify the objective existence of spatial mismatch shown in Table 1 have turned out to help the expansion of spatial mismatch theory at last (Figure 1). Employment suburbanization and residential segregation bring about the physical manifestation of residence-employment spatial separation on one hand. On the other hand, a series of factors, such as race, transportation, employment skills, employment information, etc., may create residential immobility, commuting barriers, and job searching barriers to aggravate the dysconnectivity between residence and employment. In a word, spatial mismatch is a phenomenon that reflects the loss of employment welfare due to the spatial distance of residence from employment and the lack of connectivity between residence and employment. Spatial mismatch is caused by a combination of factors such as restricted residential migration, imbalanced public transport policies, lack of individual mobility, selective discrimination in the job

market, and poor access to employment information, all of which are superimposed on the changing spatial structure of the city. Overall, research on the factors influencing spatial mismatch has evolved in two respects: firstly, from focusing on the factors influencing the distance of the relevant subject from employment (residential segregation) to the lack of connection between residence and employment (residential immobility, commuting barriers, job searching barriers); and secondly, by interconnecting and embedding these factors with space and understanding spatial mismatch from a broader perspective of spatial inequality, emphasizing the essential link between space and factors such as race, poverty, transport, and individual characteristics rather than separating them out and considering only spatial distance, so as to resolve the debate between spatial mismatch and racial mismatch, transportation mismatch, skills mismatch, information mismatch, etc.

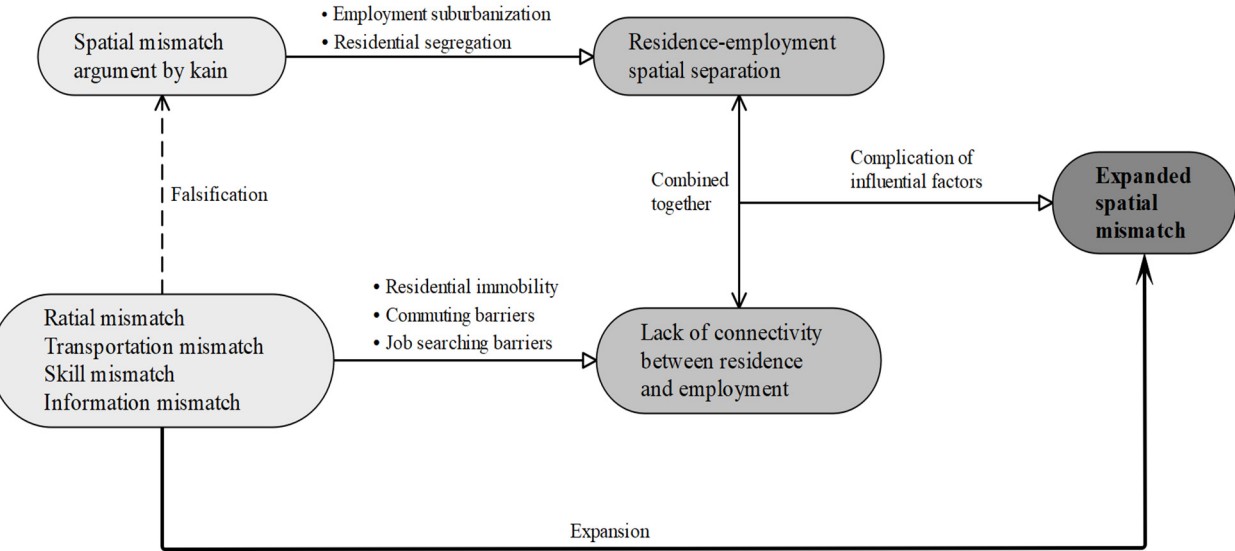

**Figure 1.** The complication of influential factors of spatial mismatch.

*3.3. Diversity in the Manifestation of Spatial Relations*

To confuse spatial mismatch with residential segregation, residential-employment separation, and lack of accessibility is in fact to confuse the measurement of spatial mismatch with the manifestation of spatial separation. No matter segregated in inner city but employment decentralized or relocated in suburbs but employment concentrated in city centers, both manifest the residence-employment spatial separation. However, spatial separation is only a pre-stage of spatial mismatch, which occurs when spatial separation results in employment loss for the group concerned. Appropriate indicators and methods to accurately measure spatial separation are the prerequisite for testing spatial mismatch. Job accessibility is an indicator commonly used to characterize the spatial relationship between residence and employment. It is literally easy to understand, but its accurate measurement is very difficult. Over the years, research has developed a number of indicators to measure spatial relationships. These indicators can be divided into two main categories: those that indirectly reflect job accessibility and those that directly reflect job accessibility (Table 2).

**Table 2.** Category of spatial relationship measurement indicators.

| Categories | Specific Indicators |
| --- | --- |
| Indirectly reflecting job accessibility | Residential Segregation<br>Residential suburbanization<br>Employment suburbanization |
| Directly reflecting job accessibility | Commuting-based job accessibility: commuting time, commuting distance, commuting costs, etc.<br>Employment-based job accessibility: ratio of job to residence |

### 3.3.1. Indirect Indicator of Job Accessibility

(1)    Residential segregation

Using residential segregation to reflect the separation of Blacks from employment was the indicator originally used in the spatial mismatch test. This is related to the initial emphasis on racial discrimination in suburban housing markets and the resulting residential segregation leading to impaired employment status for Blacks [4,5]. The logic behind this was that residential segregation would prevent the group concerned from being closer to employment growth opportunities and therefore affect their employment status. However, the degree of residential segregation can indirectly measure the separation from employment and can be seen as a crude institute for job accessibility, as residential segregation does not show the relationship between residential area and job distribution and can lead to biased results.

(2)    Residential suburbanization

Residential suburbanization is also an indirect way of reflecting the job accessibility [11,65]. The logic behind it is that inner-city Blacks are less accessible to job opportunities than suburban Blacks, and the lack of job opportunities can lead to higher unemployment, longer commuting distances, and lower wages. Earlier studies have examined spatial mismatch by comparing differences in employment participation, commuting, and wages between inner-city Blacks and suburban Blacks, but the findings vary considerably. This may be related to the simple city-suburban ring dichotomy: employment opportunities are unequal within the simple delineation of city-suburban areas, and the boundaries of city-suburban areas are constantly changing and blurring over time.

(3)    Employment suburbanization

The degree of employment suburbanization is usually measured as a ratio of the number of jobs outside the inner city to the total number of jobs. The spatial mismatch can be tested by comparing the differences in the number of jobs in inner city and suburban areas and linking them to differences in the income status of Black in inner cities and suburbs [21]. However, some studies have suggested that discussing all job numbers in general would lead to bias, and that the proportion of certain jobs suitable for certain groups should be used, such as the proportion of manufacturing, retail, and service jobs suitable for Blacks [38]. It is important to notice that using this indicator is still subject to the simple city-suburban ring dichotomy and that wage earnings will vary according to the profitability, labor market supply and demand conditions of different zones, which need to be carefully excluded from the study.

### 3.3.2. Direct Indicator of Job Accessibility

While the first three indicators have their own use logic, they can only indirectly reflect job accessibility to characterize the spatial relationship between residence and employment. Some studies have suggested the need to directly measure job accessibility, i.e., to directly reflect the proximity of the employed population to job opportunities.

(1)    Commuting-based job accessibility, such as commuting time

If the accessibility of jobs is low for a particular group, a longer commute is required for employment. While Ellwood argues that commute time is the best direct indicator of job accessibility, two contrasting conclusions have been reached by testing the relationship between commuting time and Black youth employment, with some studies that concluded no significant effect [7] and others that concluded that there was a significant effect [34,35]. The differences in empirical results may be related to sample selection bias, endogeneity of residential choices, and differences in commuting distances, commuting tools, and commuting nodes. Studies have also suggested that accessibility is a concept that involves physical, psychological, economic and social features, which makes the measurement of it invalid using traditionally place-based accessibility related to the costs needed to reach a destination [66]. Thus, explicit indicators, such as surface for vehicular traffic only,

pedestrian accesses, percentage of accesses without architectural barriers, etc., are used to measure the commute-based accessibility directly [67,68].

(2)    Employment-based job accessibility, such as employment to residence ratio

Direct measures of job accessibility can also be made by directly reflecting the matching of potential employers of different geographical units with potential jobs in the vicinity, i.e., employment to residence ratio [69]. However, a range of methodological challenges may also affect the accurate measurement of job accessibility, including measurement bias issues using all jobs rather than vacant jobs, labor market segmentation issues using all jobs without distinguishing jobs suitable for low-skilled workers, including the failure to consider job market competition from residents of neighboring communities seeking jobs in their own community and from residents of their own community working in other communities, as well as the failure to consider issues such as distance decay effects due to the spatial mobility of employed people [70]. In this regard, some studies have created functions that express job vacancies, labor market segmentation, employment competition, and the degree of spatial obstruction between two locations in order to measure spatial accessibility of jobs more accurately [71–74]. The scale is also a key issue to be considered in the accessibility measurement, since the level of accessibility of cities or metropolitan differs greatly with that of regional, local, and individual [75]. Accessibility of 15-min neighborhood level is often used responding to the current urban restructuring that much more employment activities are supposed to be done in the 15-min neighborhood instead of in city scale. There is a higher accessibility if it guarantees an adequate supply of jobs and basic services within the 15-min neighborhood [76].

### 3.4. Extension of Consequential Effects

It is always emphasized that spatial mismatch is formed when spatial separation of employment from residence and the resulting adverse labor market outcomes occurs in both stages. Spatial mismatch was first proposed through research which found that inner-city Blacks had higher unemployment rates due to geospatial mismatch from employment resources. As the research progressed, indicators of labor market outcome status such as employment participation, commuting, and income, as the main manifestations of the consequences of spatial separation, gradually became important indicators of the consequences of spatial separation in the test of spatial mismatch. Employment participation is the most commonly used indicator to characterize labor market outcomes, and where job accessibility is high, employment rates are relatively high [77,78]. Commuting distance, time, and cost were also another indicator of early empirical evidence of spatial mismatch. It was later suggested that distance, time, and cost alone do not fully represent the burden imposed on commuting, and that a combination of commuting costs may be a better option, taking into account the degree of inaccessibility and other unforeseen problems in commuting [79]. The wage income indicator was developed as a result of research that found that Black Americans in inner cities may shift from higher-wage jobs to lower-wage jobs in response to the spatial mismatch caused by suburbanization of employment, and that these Blacks may have no change in employment participation and a shorter commute, but significantly lower wages and earnings [80]. When conducting spatial mismatch tests, there are multiple combinations of how indicators are selected to reflect spatial separation as well as reflect labor market outcomes (Figure 2). This combination of different indicators, combined with the methodological challenges faced in the use of the indicators, has led to differences in the empirical results and the debate on the objective existence of spatial mismatch [81]. However, integration with labor market outcomes has been emphasized by spatial mismatch since its inception, and this is exactly where it distinguishes from macro-spatial pattern studies.

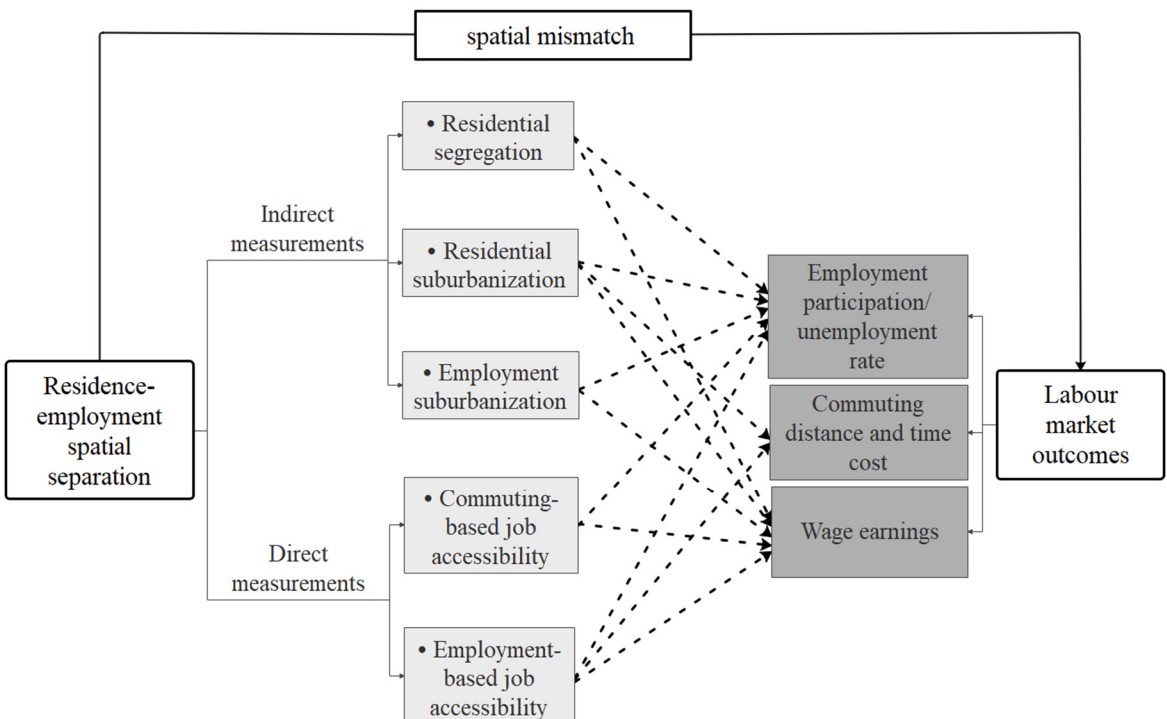

**Figure 2.** The measure of Job Accessibility and Spatial Mismatch.

In addition to impaired employment well-being resulting from the separation of residential and employment resources, residential misalignment with other spatial resources may also lead to a range of spatial welfare losses; the latter has become a powerful direction for the expansion of spatial mismatch research. The groundbreaking research in this area continues to be produced by Kain, who in 2004 raised concerns about the mismatch between disadvantaged groups and public service resources such as education and the resulting disadvantages. By using the example of public schools in Texas, he found that much of the Black-White achievement gap could be attributed to the continued residential segregation of Blacks in the inner city and the spatial inaccessibility of highly productive schools [40]. Public service resources include shopping and commercial services, health care, educational and cultural services, sports and green spaces, etc. The accessibility of public service resources is unequal across communities and is clearly linked to residential location [82]. With the suburbanization in the United States, there is a clear tendency to allocate public service resources to higher-income suburban communities. In China, the accessibility of public green space to different social classes varies and the excessive daily commute to kindergartens leads to a residence-kindergarten spatial mismatch [83]. In the UK, the accessibility of primary healthcare provision needs to be improved [84]. These studies have expanded spatial mismatch from the spatial relationship between employment and residence to public service provision. However, there is a need to further deepen the theory of spatial mismatch to guide how to bridge the gap from accessibility to public service resources to loss of public service benefits.

In other words, when spatial mismatch was first proposed, it was mainly focused on the spatial separation of residence and employment and the loss of employment welfare from the perspective of labor economics. Subsequently, most studies in the US, the UK, and other western countries also followed this line of thought, emphasizing the labor market consequences as a level that must be attended to in the study of spatial mismatch. However, with the intervention of disciplinary fields such as sociology and planning, spatial mismatch is not only used to express the separation of the spatial relationship between residence and employment, but the spatial relationship between residence and public service resources has also entered the realm of spatial mismatch research. Spatial mismatch can be used to

express the spatial relationship between multiple subjects, can be applied to areas other than the labor market, such as the study of the spatial relationship between residential areas and public service resources and the resulting loss of spatial welfare. This can dovetail with current hot issues such as spatial equality and social justice, thus continuously enrich and deepen the connotation of spatial mismatch and maintaining the vividness and lasting power of the theoretical value of spatial mismatch.

## 4. Study on the Mechanism of Spatial Mismatch: From Hypothesis to Theory

In the face of the intricate empirical research methods and results, some scholars have emphasized the importance and necessity of conducting normative research on spatial mismatch at the theoretical level [13–15]. However, it was not until the late 1990s that theoretical models related to spatial mismatch began to emerge. At the present, there are two main types of normative studies explaining the mechanism of spatial mismatch: the labor market effect model constructed from the perspective of commuting, and the search-match model constructed from the perspective of job search.

### 4.1. Description of Labor Market Effect Model

The fact that commuting may discourage the unemployed from searching for jobs in distant zones or accepting long-distance jobs (low expected net wages after removing commuting costs) has long been a major concern of empirical studies of spatial mismatch, but theoretical studies of the commuting perspective have emerged relatively late. Coulson et al. (2001) constructed a basic version of a spatial mismatch model explained in terms of commuting, where differences in the entry costs of firms in different locations and differences in the commuting ability of potential workers lead to different spatial frictions in the search matching process between potential workers and firms and ultimately to equilibrium [85]. There are more vacancies in the suburbs than in the inner city, suburban dwellers work mainly in the suburbs, some inner-city dwellers who endure long commutes work in the suburbs, and inner-city dwellers are more likely to have reverse commutes, lower wages, and higher unemployment rates than suburban dwellers. Subsequent studies have taken into account housing market discrimination and depicted the equilibrium pattern of urban structure under employment suburbanization and housing market distortions: housing market discrimination confines Blacks to living in the inner city and discourages some Blacks from accepting suburban jobs, making the labor pool of inner city larger than the suburban, resulting in impaired welfare for Blacks at both fixed-wages and flexible-wages [14]. The model does not explain the effect of spatial mismatch on the unemployment rate. Subsequent studies have solved for the effects of employment suburbanization and housing market distortions on Black unemployment by assuming that labor demand is exogenous to the wage level: Black labor tends to be supplied in the inner city regardless of the minimum-wage setting or the efficiency-wage setting, resulting in higher black unemployment in the inner city than in the suburbs [86].

### 4.2. Description of Search-Match Model

The job search process can reflect the spatial relationship between residence and employment, and the spatial distance between residence and employment may manifest itself in negative effects in terms of access to job information, job search intensity, and job search costs. Distance may reduce access to information about job opportunities, which may reduce the efficiency of job search and the likelihood of successful employment. When the expected benefits of search for potential workers are lower than the difference in commuting costs between workers and the unemployed, a spatial mismatch equilibrium tends to develop. Unemployed people living away from employment centers have lower search efficiency and higher unemployment rates [87]. Living away from the employment center, social networks are less effective in disseminating and sharing employment information, and search efficiency may be further reduced, and unemployment may be further increased [88,89]. Distance may reduce job search motivation and job search intensity,

which in turn affects employment participation levels: employment participants who live away from employment centers are less stressed about housing, less motivated to search for jobs, and do not search as frequently, increasing their short-term utility but losing long-term benefits, and may lower long-term employment possibilities and wage incomes. Spatial mismatch may also be the result of active behavioral choices by labor market participants [90]. High search costs may discourage inner-city residents from searching jobs in job-intensive suburban areas, thereby affecting employment participation levels: inner-city residents weigh search efficiency against search costs to choose their search area, and when search costs are too high in suburban areas, inner-city residents lose their incentive to search for jobs in suburban areas and unemployment increases; at the same time, high search costs reduce the bargaining ability of inner-city residents, making inner city residents bargain for lower wages in the suburbs than suburban residents [91].

Normative research can clarify the mechanism of spatial mismatch and sort out the messy results of spatial mismatch empirical studies, so as to help spatial mismatch hypothesis rise into spatial mismatch theory [92]. However, normative research has emerged late, and these theoretical models are mostly after, rather than ahead of, empirical research, resulting in a certain disconnect between normative and empirical research. For example, the exploration of factors influencing spatial mismatch in empirical research has transitioned from emphasizing residential segregation to focusing on barriers to residential migration, but so far there is a lack of normative research conducted from the factors of barriers to residential migration.

## 5. Conclusions and Discussions

### 5.1. Conclusions

Through the process of spatial mismatch from "hypothesis formulation, objective existence debate, theoretical connotation expansion, hypothesis into theory", it is found that the process of spatial mismatch research development is essentially a process of continuous re-conceptualization of spatial mismatch theory. Kain's groundbreaking research focused on the adverse effects of space and distance on the employment of Blacks in inner cities in the United States, which was later described by scholars as the spatial mismatch hypothesis when distilling Kain's findings, resulting in a rather clear but narrow definition of spatial mismatch in the early years. The initial definition of spatial mismatch was too narrow, which led to a long debate on the objective existence of spatial mismatch. As the understanding of spatial mismatch deepens, research is no longer limited to testing spatial mismatch and debating the objective existence of spatial mismatch but begins to enrich and expand the connotation of spatial mismatch at different element levels. This has led to a general understanding of the current connotation of spatial mismatch, namely that the misalignment or inconsistency between residence and the location of job opportunities combined a series of geospatial barriers to employment together results in the disadvantageous position of the relevant subject in the labor market, describing the spatial relationship between residence and employment, focusing on the resulting spatial barriers and the labor market disadvantage of the relevant subject, with the complex factors that trigger spatial mismatch not explicitly specified but implied.

Residential segregation and poor transportation may also lead to a misalignment between residence and the location of public service resources, creating spatial barriers to access and enjoyment of public service resources by the relevant groups, reducing the accessibility of public service resources, and bringing about a loss of public service welfare for the relevant subjects. The use of spatial mismatch to express the spatial separation between residential areas and public service resources and the resulting loss of welfare makes spatial mismatch more possible. Spatial mismatch is a comprehensive reflection of the unequal spatial opportunities in accessing employment resources and public service resources such as education, health care, and green spaces, and its essence is to reveal the survival and living conditions of the relevant subjects in the changing spatial structure of the city from the perspective of the spatial separation between the place of residence and the

relevant resources. It is in the ongoing debate on the objective existence of spatial mismatch, in the practice of focusing on a wider group of people concerned, in the continuous improvement of the method of measuring spatial relations, and in the attempt to apply spatial mismatch to areas other than the labor market that help to give the theory of spatial mismatch deeper connotations, making it still a living theoretical value today (Figure 3).

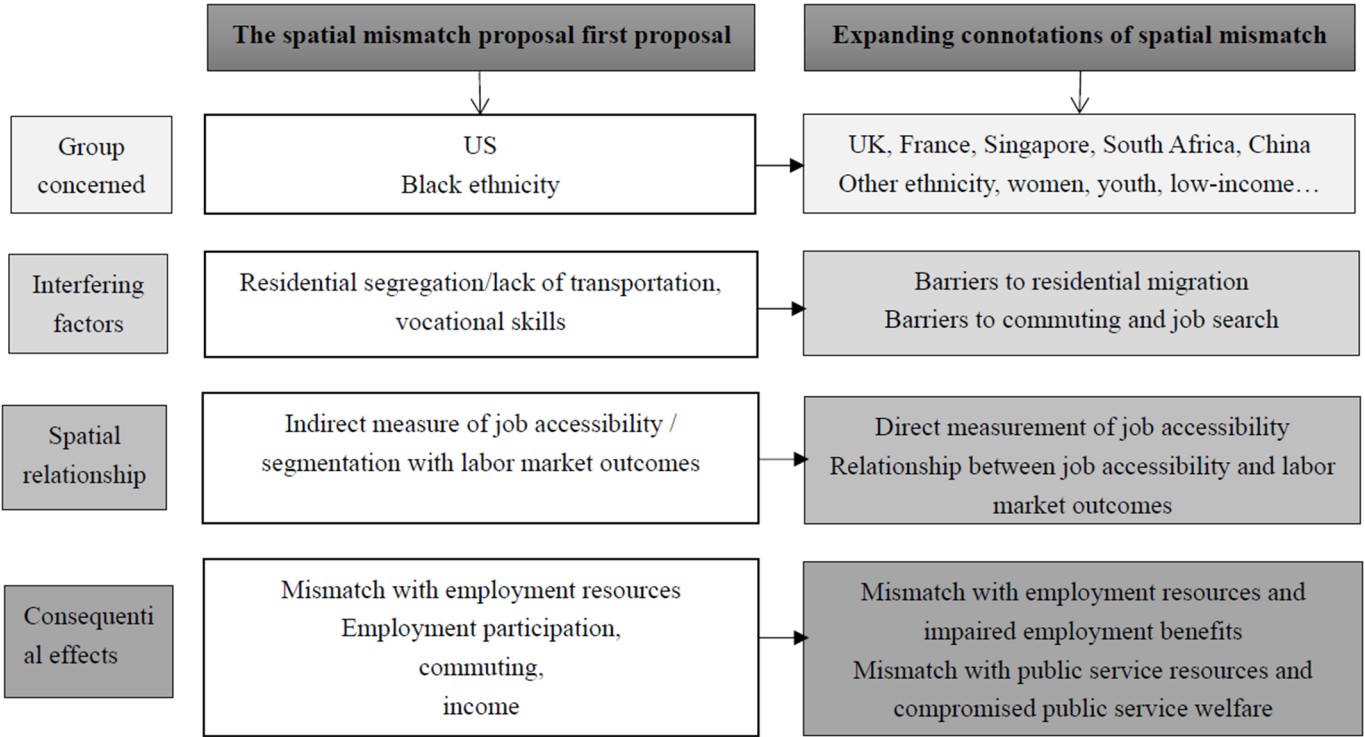

**Figure 3.** The Connotative Expansion of Spatial Mismatch Theory.

*5.2. Discussions, Limitations and Future Research Directions*

The evolution and expansion of the connotations of spatial mismatch theory did not happen overnight. Some are still at a discovery stage, such as in the accessibility of public service resources and the resulting welfare-consequential effects. Some have yet to be examined in depth, such as action mechanisms that lead to the separation of residence from employment and prevent the group concerned from coping with this separation through migration. Some have not been fully applied, such as the improved indicators for measuring job accessibility to overcome methodological challenges and the test method for spatial mismatch. Under the current trends of "urban restructuring and spatial separation have become inevitable", "the pursuit of social equality and spatial justice", and "reflecting the differences of subjects and caring for humanistic demands", it is a challenge and a future direction for spatial mismatch research to focus on the welfare loss of the relevant subjects from the perspective of spatial mismatch and analyze the deep-seated causes and mechanisms of spatial mismatch.

As for limitations, firstly, the spatial mismatch theory itself has been detailed analyzed to help better understand the roots of the spatial mismatch theory and its potential and next evolutions. However, actions and interventions and policy implementations that oriented to edit and improve the spatial mismatch have not been paid enough attention. Therefore, this article is useful for scholars. While for city planners and administrators, it can provide theoretical preparation but cannot provide practical methods and policies about how to design and deliver services that are easily and conveniently available. Secondly, spatial mismatch theory reveals the spatial separation and the consequential adverse effect on the group concerned, but how the group concerned will react to such adverse effect have not been included in this research. Since the possible residential, employment, and travel

actions of the group concerned draw main references to the governors, the miss of possible action studies may also decrease the practical value of this research to some extent. Anyway, the specific practice for overcoming spatial mismatch of different countries differs since the influential factors and spatial separation manifestations are inconsistent. In addition, taking considering that, focusing on the intervention policies of spatial mismatch may dilute the live of logic about analyzing the hierarchical elements of spatial mismatch, this acritical have studied spatial mismatch theoretically, not applicably.

The spatial mismatch in the United States stems from the suburbanization of metropolitan employment and the segregation of the housing market in the late 1960s, manifesting itself in long commutes to suburban employment while living in the inner city. The spatial mismatch in many countries outside the United States, on the other hand, stems mainly from the transformation of the economic system and changes in the spatial structure of the city, manifesting itself in the concentration of employment in the central city and living in the urban periphery. Although the spatial mismatch in various countries is quite different in terms of its background and manifestation, its essence is always due to the spatial separation between residence and employment, resulting in barriers to employment and income for the subjects concerned. The current widespread urban expansion, renewal, and transformation in developing countries have inevitably led to the spatial separation of residence-employment and negatively impacted the employment welfare of the relevant subjects, and it is urgent to analyze and explore possible solutions from the perspective of spatial mismatch. The measurement of the spatial relationship between residence and employment cannot be confined to the macro level of aggregation. It is necessary to closely associate spatial separation with labor market outcomes, analyze the impact of spatial separation of residence and employment on the labor market, and then extend it to areas such as public service provision. In this way, the essence of spatial mismatch theory is grasped, and the scope of its application is expanded, so that it continues to maintain its vivid theoretical and applied value.

**Author Contributions:** Conceptualization, L.W. and C.W.; formal analysis, L.W. and S.Z.; writing—original draft preparation, L.W. and S.Z.; writing—review and editing, L.W. and C.W. All authors have read and agreed to the published version of the manuscript.

**Funding:** This research was funded by the Humanity and Social Science Youth Foundation of Ministry of Education of China, Grant Number 20YJCZH158, the Youth Innovative Talents Project of General Colleges and Universities in Guangdong, China, Grant Number 2018WQNCX037, and the Special Research Project of Philosophy and Social Sciences in Colleges and Universities in the 13th five-year plan of Educational Science in 2019, Guangdong, China, Grant Number 2019GXJK057.

**Institutional Review Board Statement:** Not applicable.

**Informed Consent Statement:** Not applicable.

**Data Availability Statement:** Not applicable.

**Conflicts of Interest:** The authors declare no conflict of interest.

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
