# Peer review of "A Review of Spatial Mismatch Research: Empirical Debate, Theoretical Evolution and Connotation Expansion"

_land, doi:10.3390/land11071049_

Round 1

Reviewer 1 Report

The spatial mismatch theory can be useful and interesting for scholars, due to its links to the actual climate, environment and mobility needs affecting territorial and urban systems.

So, first of all, the authors could make the effort to consider this aspect within section 1. In line with this perspective, the main conclusions of previous works summarised in table 1 could be related to the aspects described in section 3. In this way, the authors could strengthen the scientific soundness of their work, as the links between the consolidated elements of the spatial mismatch theory and its potential developments would be better underlined.  These new links could then added to figure 3.

Within section1, first and section3, then, wider greater emphasis on accessibility issue need to be provided. The authors refer to spatial relations, job accessibility, distances, etc. which are all elements related to accessibility. Therefore, follows are some useful references:

Guida, C., & Caglioni, M. (2020). Urban accessibility: the paradox, the paradigms and the measures. A scientific review. TeMA-Journal of Land Use, Mobility and Environment13(2), 149-168.

Kaplan, N., Burg, D., & Omer, I. (2022). Multiscale accessibility and urban performance. Environment and Planning B: Urban Analytics and City Science49(2), 687-703.

Gaglione, F., Zucaro, C. G. F., & Cottrill, C. 15-minute neighbourhood accessibility: a comparison between Naples and London. European Transport 85(5). https://doi.org/10.48295/ET.2021.85.5

Pellicelli, G., Rossetti, S., Caselli, B., & Zazzi, M. (2022). Urban regeneration to enhance sustainable mobility. TeMA-Journal of Land Use, Mobility and Environment, 57-70.

Following some further overall comments on the paper " A review of spatial mismatch research: Empirical debate, Theo- 2 retical evolution and Connotation Expansion ". The subject of the spatial mismatch can contribute to broadening the scientific debate on issues related to accessibility and to the other current and next future challenges for urban and territorial systems. Spatial effects of actions and interventions oriented to edit and improve the built environment are useful for scholars but moreover for technicians and decision-makers that have to govern a territory. From this perspective, the empirical work made by authors can help to better understand the roots of the spatial mismatch theory and its potential and next evolutions. The references provided allow to draw an appropriate state of art, even if further references to the linked topics need to be added. The authors provided a scientific review of this issue and so they should better frame the topic in a wider debate related to the linked issues of spatial mismatch. Otherwise, the risk is that the readers cannot grasp the strengths of this theory and its usability within the scientific practice. 

It is a qualitative work (a literature review mainly) so an evaluation of the method is not applicable. On the other hand, I can assert that the paper is well structured and that the authors tried to reach their aims by a rigorous qualitative research path.

Author Response

We really appreciate all constructive comments and suggestions from reviewers. Your helps and supports are the important motivation for us to improve the manuscript. We made the revisions based on each comment and suggestion, we also submit a revision note to explain how we revised.

Reviewer 2 Report

Dear Authors

thank you very much for sending your paper to review. The paper is well prepared, the structure is fine and the literature review was made with paying attention to details. In my opinion, the paper is publishable if considering the improvements mentioned below. They are both theoretical and technical, in order of appearance:

1) Please revise lines 97-98
2) Please revise the text in order to avoid using exclusion words talking about ethnicity
3) Please consider adding the term "urban sprawl" for uncontrolled suburbanisation in the Introduction or part 2
4) Revise the line 370
5) Please rename the 4.1. and 4.2. sections - model can be graphical, mathematical - this description you provided is not a model or clarify please what kind of model it is - maybe this is a description of possible models? I would advise you to change it.
6) Please add 5.3. Limitations and future research directions and summarize the shortcomings of your research and propose future directions regarding those limitations and your results.

Author Response

(The authors gave the same response as above.)

Reviewer 3 Report

This paper is an excellent theoretical synthesis on spatial mismatch research. It has great theoretical rigor and, in my opinion, can be published without changes. Congratulations.

Author Response

(The authors gave the same response as above.)
